# Decision Conflicts in Clinical Care during COVID-19: A Patient Perspective

**DOI:** 10.3390/healthcare10061019

**Published:** 2022-05-31

**Authors:** Jörg Haier, Johannes Beller, Kristina Adorjan, Stefan Bleich, Moritz De Greck, Frank Griesinger, Alexander Hein, René Hurlemann, Sören Torge Mees, Alexandra Philipsen, Gernot Rohde, Georgia Schilling, Karolin Trautmann, Stephanie E. Combs, Siegfried Geyer, Jürgen Schäfers

**Affiliations:** 1Comprehensive Cancer Center Hannover, Hannover Medical School, 30625 Hannover, Germany; beller.johannes@mh-hannover.de (J.B.); schaefers.juergen@mh-hannover.de (J.S.); 2Medical Sociology Unit, Hannover Medical School, 30625 Hannover, Germany; geyer.siegfried@mh-hannover.de; 3Department of Psychiatry and Psychotherapy, University Hospital, Ludwig Maximilian University of Munich, 81377 Munich, Germany; kristina.adorjan@med.uni-muenchen.de; 4Department of Psychiatry, Social Psychiatry and Psychotherapy, Hannover Medical School, 30625 Hannover, Germany; bleich.stefan@mh-hannover.de; 5Department of Psychiatry, Psychosomatic Medicine and Psychotherapy, University Hospital, Goethe University Frankfurt am Main, 60528 Frankfurt, Germany; moritz.degreck@kgu.de; 6Department of Hematology and Oncology, Pius-Hospital Oldenburg, Carl von Ossietzky University, 26129 Oldenburg, Germany; frank.griesinger@pius-hospital.de; 7Department of Gynecology and Obstetrics, Erlangen University Hospital, 91054 Erlangen, Germany; alexander.hein@uk-erlangen.de; 8Comprehensive Cancer Center Erlangen-EMN (CCC ER-EMN), 91054 Erlangen, Germany; 9Department of Psychiatry, Carl von Ossietzky University, 26129 Oldenburg, Germany; rene.hurlemann@uol.de; 10Department of General, Visceral and Thoracic Surgery, Friedrichstadt General Hospital, 01067 Dresden, Germany; soeren-torge.mees@klinikum-dresden.de; 11Department of Psychiatry and Psychotherapy, University Hospital Bonn, 53105 Bonn, Germany; alexandra.philipsen@ukbonn.de; 12Department of Respiratory Medicine and Allergology, University Hospital, Goethe University, 60590 Frankfurt am Main, Germany; gernot.rohde@kgu.de; 13Department of Hematology, Oncology, Palliative Care and Rheumatology, Asklepios Hospital Altona, Asklepios Tumorzentrum, 22763 Hamburg, Germany; g.schilling@asklepios.com; 14Department of Hematology and Oncology, University Hospital Carl Gustav Carus, 01307 Dresden, Germany; karolin.trautmann@uniklinikum-dresden.de; 15Department of Radiation Oncology, Klinikum rechts der Isar, Technical University of Munich, 81675 Munich, Germany; stephanie.combs@tum.de

**Keywords:** decision conflicts, moral distress, uncertainty, oncology, psychiatry, COVID-19

## Abstract

(1) Background: Uncertainty is typical for a pandemic or similar healthcare crisis. This affects patients with resulting decisional conflicts and disturbed shared decision making during their treatment occurring to a very different extent. Sociodemographic factors and the individual perception of pandemic-related problems likely determine this decisional dilemma for patients and can characterize vulnerable groups with special susceptibility for decisional problems and related consequences. (2) Methods: Cross-sectional data from the OnCoVID questionnaire study were used involving 540 patients from 11 participating institutions covering all major regions in Germany. Participants were actively involved in clinical treatment in oncology or psychiatry during the COVID-19 pandemic. Questionnaires covered five decision dimensions (conflicts and uncertainty, resources, risk perception, perception of consequences for clinical processes, perception of consequences for patients) and very basic demographic data (age, gender, stage of treatment and educational background). Decision uncertainties and distress were operationalized using equidistant five-point scales. Data analysis was performed using descriptive and various multivariate approaches. (3) Results: A total of 11.5% of all patients described intensive uncertainty in their clinical decisions that was significantly correlated with anxiety, depression, loneliness and stress. Younger and female patients and those of higher educational status and treatment stage had the highest values for these stressors (*p* < 0.001). Only 15.3% of the patients (14.9% oncology, 16.2% psychiatry; *p* = 0.021) considered the additional risk of COVID-19 infections as very important for their disease-related decisions. Regression analysis identified determinants for patients at risk of a decisional dilemma, including information availability, educational level, age group and requirement of treatment decision making. (4) Conclusions: In patients, the COVID-19 pandemic induced specific decisional uncertainty and distress accompanied by intensified stress and psychological disturbances. Determinants of specific vulnerability were related to female sex, younger age, education level, disease stages and perception of pandemic-related treatment modifications, whereas availability of sufficient pandemic-related information prevented these problems. The most important decisional criteria for patients under these conditions were expected side effects/complications and treatment responses.

## 1. Introduction

Uncertainty is inherent in sudden viral outbreaks, such as the current COVID-19 pandemic, or similar healthcare crises [1]. It does not only affect caregivers and healthcare politicians, but also to a large extent patients. During this crisis, especially in the early phase with a large deficiency of clinical evidence, it was therefore vital to apply ethical perspectives in the clinical practice to handle this uncertainty and avoid decisional conflicts [2]. In addition, the pandemic affects availability and accessibility of healthcare very intensively, inducing moral distress for healthcare providers [3] and decisional problems for patients. For example, during the COVID-19 pandemic patients and caregivers experienced delays [4] and treatment modifications in various clinical settings [5,6] that negatively affected their psychological wellbeing and caused decisional conflicts. Shared decision making (SDM) rapidly became challenging during the pandemic, and affected various participants in healthcare processes to a different extent. For example, decisional conflicts were reported for patients undergoing surgery [7] and their extent was related to sociodemographic factors such as race. In this context, SDM was addressed early as ethical dilemma that may affect the emotional health status of healthcare providers and patients [8,9].

As a consequence, the pandemic induced a need for adapted criteria for SDM processes [10], but investigations aiming at patients’ attitudes towards SDM and predictors of decisional preferences during the pandemic are very rare [11,12]. For example, qualitative analyses by Edge et al. [13] and Butow et al. [14] showed pandemic-related SDM vulnerability of specific patient groups, such as in oncology, characterized by psychological distress, fear of virus susceptibility, practical issues in daily life, disruptions to treatment and services, information needs, and caregiver issues. In addition, various aspects of uncertainty as stressors for patients were identified [15]. This psychosocial impact seems to be characteristic for patient subgroups, such as with different age [16] or comorbidities [17]. Furthermore, this subjective distress in response to COVID-19 has a dynamic perception during the pandemic [18]. The investigation of distress has been focused towards its prevalence and changes during the pandemic. Decisional uncertainty, decisional conflicts and specific factors of the pandemic that determine the decisional dilemma under the specific conditions of COVID-19 are mainly missing. Since its understanding appears to be a prerequisite for adapted SDM processes, identification of vulnerable subgroups and their determinants is required.

Although the vast majority of the currently published investigations targeting decisional uncertainty and conflicts in patients are related to cancer, the assumption of a general challenge for many entities seems to be reasonable. In our investigation, we aimed at the characterization of decisional uncertainty and conflicts in patients during the first phase of the pandemic throughout Germany. We hypothesized that sociodemographic factors and the individual perception of pandemic-related problems are determinants of the perception of a decisional dilemma for patients. We chose cancer and psychiatry patients as two groups that were assumed especially vulnerable for those problems. To obtain a nationwide picture and to include the very different extent of COVID-19 throughout Germany in this pandemic phase, a large number of study sites in all areas of the country were included. For identification of determinants and related interlinked dependencies we chose a survey and multivariate analytical approach.

## 2. Materials and Methods

### 2.1. Questionnaire Development

Questionnaires for acquisition of various stakeholder perspectives regarding pandemic-related decisional uncertainties and related distress were developed as previously described [19]. Briefly, the questionnaires were covering 5 decision dimensions (conflicts and uncertainty, resources, risk perception, perception of consequences for clinical processes, perception of consequences for patients). Every dimension included 3–5 questions, some of them with questions on detailed aspects of the topics covered. OnCoVID questionnaires were validated involving 5 representatives of each professional group and patient representatives. Out of the 216 different questionnaire items [3] 108 variables were related to patient target groups and 24 items covering oncology and psychiatry patients were included in this investigation.

### 2.2. Sample

Cross-sectional data from the OnCoVID trial (ethical approval 9199_BO_K_2020) were used. Data were collected by pen-and-paper survey between October 2020 and June 2021 from 540 patients (283 females, 245 males, 12 N/A) in 11 participating hospitals (university and nonacademic hospitals) throughout Germany. Patients were contacted via the cooperating clinics and outpatient centers and invited by mail to participate in the survey. Participating patients were either actively treated in oncology and psychiatry departments on an inpatient or outpatient basis during the pandemic. Questionnaires included only very basic demographic data (age, gender, stage of treatment and educational background) that were previously identified as potential factors for decisional burden in a quantitative presetting.

### 2.3. Variables

Decision uncertainties and distress were operationalized using 5-point scales that were adapted according to the related questions (from “not at all” to “completely”; “not at all/seldom” to “most of the time”; “much less” over “no changes” to “much more”; “not likely” to “very likely” “very negatively” over “no changes” to “very positively”). A 4-point scale (“Not at all /seldom”, “Sometimes”, “Frequent”, “Most of the time”) captured questions related to the frequency of occurrence, such as for frequency of anxiety, stress, etc., during the last two weeks. All these scales can be considered as equidistant. Additional demographic variables contained gender (male, female), specialty (psychiatry, oncology), age (years), stage of treatment (“initial treatment after diagnosis”, “treatment continuation”, “recurrence/metastasis/crisis treatment”, “follow up”) and educational background (7 categories). Dichotomic answers were coded as “yes” or “no”.

### 2.4. Data Analysis

Decisional uncertainty and distress as well as items reflecting the psychological environment of the patients were first analyzed by descriptive statistics using histograms and boxplots. Items were characterized by mean ± SD and 95% confidence intervals.

ANOVA and Tukey-HSD for post hoc tests were applied to compare patient groups and ordinal variables. Pearson rank correlation was used for comparison of two groups. For continuous variables, *t*-tests were applied. Chi^2^-tests with continuity correction were performed in the case of categorical variables.

For items with high similarity, multivariate factorial analysis was performed as principle component analysis (PCA). The parameters were combined in a stepwise approach and a sufficient number of significant correlations was approved by Kaiser–Meyer–Olkin criterion (KMO accepted if >0.5) and significance of Bartlett test for sphericity.

For multivariate analysis, a two-step approach was applied. First, a nominal regression analysis was performed. The identified parameters were used for a subsequent classification-tree analysis. As dependent variables, “Decisional Uncertainty” and “Decisional Distress” were defined. Nonrespondents were excluded pairwise from analyses of the respective items. As build-up method, the Chi^2^ automatic interaction detection (CHAID) was used. The number of levels was limited to *n* = 4 and the minimum size of knots was determined as *n* = 50 participants. Significance for splitting was accepted for *p* < 0.005.

All analyses were performed using SPSS26.

## 3. Results

### 3.1. Questionnaire Response

Overall, *n* = 540 (730 female, 473 male, 9 N/A) were returned (response rate of 54.8%). Average age of participants was 54.7 ± 16.9 years (range 15–88 years). Patients in oncology (60.7 ± 13.7 years) were significantly (*p* < 0.001) older than in psychiatry (39.9 ± 15.0 years). Only 32 psychiatry patients (20.8%) had experienced COVID-19 quarantine before the data collection. Within the patient group, 33.7% had initial treatment, 30.0% continued previous treatment, 22.2% were treated due to recurrence/progression of disease and 7.8% were in follow-up (6.3% N/A) at the time of questionnaire. In oncology, significantly more patients were in acute treatment, whereas psychiatry patients were mainly continuously treated (*p* < 0.001). School education was distributed as 115 (21.3%) lower, 185 intermediate (34.3%) and 189 higher school levels (44.4%) (not different between the entities). 

### 3.2. Decisional Uncertainty and Resulting Decisional Distress

11.5% of all patients described intensive uncertainty in their clinical decisions (answering options “A lot” and “Completely”). The extent of this perception was significantly different between oncology and psychiatry patients (*p* < 0.001). (Figure 1A) This resulted in high decisional distress in 17.4% (oncology) and 32.9% (psychiatry). (Figure 1B) As expected, reflections of decisional uncertainty and resulting distress during decision making correlated highly significantly (R^2^ = 0.64; *p* < 0.001) if this was related to their own treatment decision. General pandemic-related decisional uncertainty was correlated with the distress levels to a lesser extent, but was still significant (R^2^ = 0.30; *p* < 0.001). Overall, this general decisional uncertainty was significantly less reported than uncertainty related to own treatment decisions of the patients. Resulting distress due to treatment-related uncertainty showed the highest values (*p* < 0.001). For the entire patient group, a significant influence of the stage of the disease was found (*p* = 0.005), but this was lost if oncology and psychiatry patients were analyzed separately.

In the next step, we evaluated the requirement for treatment modifications due to the pandemic from the patients’ perspectives. Overall, 13.0% reported extensive changes (“A lot” or “Completely”); 30.1% of the psychiatry patients acknowledged intensive treatment modifications, whereas 75.3% of oncology patients answered “Not at all” for this question (*p* < 0.001) (Figure 1D).

The perception of the decisional modification requirements and uncertainty was influenced by the stage of treatment and significant differences occurred between these groups (*p* = 0.004 and *p* = 0.005), respectively). (Appendix A) Reflection of treatment modification was significantly related to the educational status for the entire group (*p* = 0.014), but this was lost within the entity groups (*p* = 0.234 and *p* = 0.093). However, differences for uncertainty between the educational groups were not found (*p* = 0.102). In a multivariate ANOVA including all uncertainty and conflict items, the educational status was only significantly related to these treatment modification requirements (*p* = 0.041) (details not shown). General uncertainty, uncertainty related to their own treatment, resulting distress, perception of own risk and requirements for treatment modifications were highly significant correlated with each other (Appendix A). Therefore, to avoid collinearity in multivariate analysis, these items were individually selected as applicable.

### 3.3. Consequences of Decisional Uncertainty and Conflicts

Patients were questioned regarding their psychological environment during the two weeks before they answered the questionnaire. High occurrence (“Frequent” or “Most of the time”) was reported in 24.2% for anxiety, 26.8% for depression, 25.4% for loneliness and 31.3% for stress. In contrast, loss of hope was found in more than half of the patients. (Figure 2A) Assessment of risk perception (“Own risk of patients during pandemic”) also showed 15.7% with high rating values (“A lot” or “Completely”) (Figure 2B).

Decisional uncertainty and distress were highly significant correlated with anxiety, depression, loneliness and stress. Much lower correlation was found with perception of own risk and hope. (Appendix A) Interestingly, perception of own risk was associated with anxiety, and to a lower extent with depression, loneliness and stress, but not with loss of hope. In all five items, psychiatry patients reported on average significantly worse values than oncology patients (*p* < 0.001). For aggregation of the items of anxiety, depression, loneliness, stress and hope, PCA-based factorial analysis generated a single factor for patients’ individual psychological status. High KMO (0.792) and significant Bartlett test (*p* < 0.001) demonstrated high reliability of this obtained parameter, which was used for further analyses. (Appendix A) This aggregated factor differed significantly between the age groups, the specialty and the stage of the disease. Younger and female patients, patients treated for recurrence/metastasis/crisis and those in psychiatry had the highest factorial values (*p* < 0.001), whereas the educational status showed only a slight trend (*p* = 0.231) (Appendix A).

Furthermore, patients were asked which criteria were important for their treatment decisions under the conditions of the pandemic. 36.1–48.9% of them reported that specific pandemic-related criteria were not important for their decisions. In contrast, high importance (“A lot” or “Completely”) was attributed to a different extent by large subgroups to treatment response (38.7%), symptom control (37.5%) and side effects/complications (27.7%). As expected, these three items showed significant correlations between each other (*p* < 0.001), but only weak relation to the pandemic-related treatment modifications. (Appendix A) In addition, in all three categories psychiatry patients reported significantly higher values than oncology patients (data not shown). Interestingly, only 15.3% of the patients (14.9% oncology, 16.2% psychiatry; *p* = 0.021) considered the additional risk of COVID-19 infections as very important for their decisions. (Figure 2C) Similarly, 22.0% reflected this infection risk as high burden for themselves (“A lot” or “Completely”) without differences between the entity groups. (Figure 2D) The relationship to the other items was rather weak (R^2^ < 0.14).

In addition, since shared decision making, especially under pandemic conditions, requires informed patients, they were questioned whether the available information was sufficient for handling their disease during the pandemic. Low satisfaction (“Not at all” or “A little”) with information provided by their medical caretaker (29.6%) or publicly available (32.2%) was reported by larger subgroups of the patients. On the other side, 50.1% (healthcare provider information) and 40.1% (public information), respectively, considered disease-related information availability as sufficient (“A lot” or “Completely”). This was slightly negative correlated with distress and uncertainty (Figure 3).

### 3.4. Reasons and Vulnerability

As a last step, we intended to identify characteristics of the patient groups that are specifically vulnerable for having decisional uncertainty and resulting distress. In a stepwise approach, we included all items that showed promising correlations with these target parameters. However, since the items pointing towards the psychological status of the patients may act as predictors as well as consequences of the pandemic-specific decisional dilemma, the obtained factor was initially not included in the regression analyses.

Nominal regression analysis targeting “Decisional Uncertainty” resulted in a highly significant prediction model (Likelihood-Quotient Test *p* < 0.001). Pseudo-R^2^ (Nagelkerke = 0.718) also confirmed high model quality. The obtained regression coefficients were significant or nearly significant for nine parameters (Table 1A). The resulting classification showed 64.8% correct predictions at the five-point scale. If the prediction of neighbor categories was also considered as acceptable, 85.4% of the patients were sufficiently classified using these regressors (Table 1B). After adding the factor psychological status as covariate to the regression analysis, this parameter contributed significantly to the model while maintaining all other general involvements. The overall predictive fit (Nagelkerke = 0.745) and the classification (65.5% and 86.3%, respectively) improved slightly (Appendix A).

In a similar manner, “Decisional Conflicts” were evaluated. Likelihood-Quotient Test (*p* < 0.001) and Pseudo-R^2^ (Nagelkerke = 0.750) again supported high model quality. However, optimal prediction was achieved by as many as 11 parameters. Although the items “SARS Additional Risk” and “Specialty Group” had only borderline significance, their removal from the model resulted in worse classification correctness, and both items were therefore left in the final model for this target. The optimized parameter lists for both targets were only in part overlapping (Table 2A). Classification rates were 64.6% and 85.9%, respectively (Table 2B). In contrast to the uncertainty, the addition of the psychological status factor did not add predictive power, and the resulting model became worse for this target parameter (Data not shown).

## 4. Discussion

During the pandemic, larger subgroups of patients have suffered from decisional uncertainty and decisional distress that can be, at least in part, attributed to the specific conditions of COVID-19. Although the uncertainty levels in our investigation were lower in patients compared to the very recently published values for healthcare professionals [3] this perception induced high levels of decisional distress in up to one third of the patients. In our investigation we aimed to characterize these subgroups that appear to be vulnerable for the decisional dilemma under pandemic conditions in order to provide dedicated support programs for these patients, especially for sufficient SDM.

Based on an inferential analysis, we identified determinants that can explain large parts of the perception variance and characterize decisional vulnerability. Younger and female patients, patients in specific treatment situations and the perception of intensive treatment modifications due to the pandemic had a higher risk for decisional uncertainty and distress. These determinants were comparable to other investigations [20] or similarly identified as predictors for general distress [21,22]. The additional risk to obtain a SARS-infection and its resulting burden, as well as fears to keep the distance regulations during the treatment, were additional factors that worsened the decisional dilemma of the patients. This additional COVID-19 related risk along with expected treatment response and side effects/complications were decisional criteria that also intensively determined the occurrence of decisional problems. Furthermore, patients who were in acute oncological situations or required treatment due to advanced disease stages had a dedicated risk for decisional problems, which was similarly found by Lou et al. [23]. In contrast, other aspects, such as own quarantine and requirements of SARS testing, did not contribute to the explanation of the target parameters. Moreover, we previously reported that the objective pandemic load with very high differences between the regions in Germany did not affect decisional uncertainty and distress [3].

Similar patterns of the decisional issues and their determinants during the pandemic were found in our analysis in oncology and psychiatry patients, but to a very different extent. Cancer is usually considered life-threatening by patients and its treatment is frequently related to high risks for the patients. However, cancer patients might be already adapted to these high risks and decisional challenges because of the underlying disease, and additional impact by pandemic-specific aspects has lower effects on SDM in these patients [22]. In contrast, uncertainty and conflicts may be part of the underlying disease in psychiatry patients and additional triggers in this direction may rapidly worsen the decisional dilemma. Taken together, prognosis or survival impairment, such as in cancer, and perception disturbances, such as in psychiatry, might represent two ends of the spectrum of decisional impairment and the different disease-related impacts of the pandemic regarding SDM.

In our analysis, we also found factors that may have protective effects regarding the decisional burden. Satisfaction with provided external or internal information about the pandemic and its relation to the underlying disease of the patients was a strong predictor of lower uncertainty as well as distress. This important role of communication and information as coping strategy has been supported by other investigations [24,25]. Interestingly, lower school education was related to fewer decisional disturbances. This observation was similarly reported by Cona et al. [26] targeting pandemic-related risks, and by Köther et al. [11] investigating decision participation preferences.

A clear relationship between specific decisional problems induced by the pandemic and the psychological status of the patients was observed in our study. Comparable results were found by Gultekin et al. [27] reporting high depression scores and having experienced modifications of care due to the pandemic as predictors of this interrelationship. Very likely pandemic-dependent and disease-related, pandemic-independent psychological effects interfere and potentially enhance each other [17]. Patients who are already endangered regarding distress due to their underlying disease seem to have a specific additional risk to be confronted with impaired SDM opportunities.

Increased prevalences of psychological disorders during the COVID-19 pandemic and decline in psychological wellbeing have been described similarly to our results [28,29]. Other studies reported higher rates [30,31,32,33], but used instruments targeting different aspects of distress. For example, health-related quality of life (HRQOL) questionnaires, Distress Thermometer, Hospital Anxiety and Depression Scale (HADS) and Perceived Stress Scale (PSS) were used in various studies [21,24,25,34,35] capturing disease-related distress that do not address the specific pandemic-related decisional dilemma. Other investigations addressed pandemic-related topics with a focus on risk perception [36]. Therefore, these results are only in part comparable. However, the general picture of highly variable perception of pandemic effects on personal decision making by patients was similarly found in other entities, such as elective surgery, and in various countries, such as the UK [37,38].

Fear of COVID-19 or disease progression, disruption of oncology services, cancer stage and immunocompromised status were proposed to induce this psychological distress in cancer patients, which can influence patients’ decisions about treatment. To our knowledge, data on psychiatry patients have not been published yet. The importance and impact of background variables regarding the psychological distress and uncertainty is highly variable, as summarized in a recent review [39]. Based on our findings with an independency from the regional COVID-19 impact that was also seen in other German [40,41] and Italian [26,42] investigations, we conclude that these differences might be more related to the underlying healthcare systems, sociocultural influences or general residence to healthcare crisis in different countries.

The strengths of this investigation are the inclusion of a large number of centers throughout Germany representing a very different involvement in the pandemic at the time of data collection and the broad area of potential determinants for decisional burden covering five different domains. This enabled a multivariate approach for identification of their specific roles. Since we used mailing acquisition of patients that were treated during the early phase of the pandemic, we cannot completely rule out a selection bias within the respondents. However, the sufficient response rate and the relatively high number of included patients likely minimize this problem. Furthermore, although we included trial sites throughout the country, we did formally analyze for representativeness of the German population. However, the authors are convinced that the investigated patient cohort provides a general picture regarding decisional uncertainty, conflicts and burden. Furthermore, identification of the complex network of determinants of patients’ burden due to a pandemic has rarely been investigated yet, and this study provides novel evidence for the importance of investigating their specific perspective.

## 5. Conclusions

The COVID-19 pandemic induced specific decisional uncertainty and distress in patients who were accompanied by intensified stress and psychological disturbances. Determinants of specific vulnerability were related to females, younger age, education level, certain disease stages and perception of pandemic-related treatment modifications, whereas availability of sufficient pandemic-related information prevented these problems. For patients, expected side effects/complications and treatment responses were the most important decisional criteria under these conditions. In contrast, the additional risk related to SARS infections did not show relevant impact on SDM in our investigation.

Distress and uncertainty driven by the COVID-19 pandemic should be addressed as part of the early crisis management in clinical practice, and additional psychological and social support targeting specific practical and emotional problems in vulnerable groups should be provided for those patients [34].

Consequences recommended for pandemic patient management of clinical caregivers include:Assessing the level of uncertainty and decisional burden in patients, especially additional effects of the pandemic;Considering potential modifications of decisional criteria in patients due to pandemic issues;Communicating one’s own decisional uncertainty with patients to ensure shared decision making;Organizing sufficient information management, especially when evidence deficits occur;Supporting specific groups of vulnerable patients in handling pandemic-related additional burden.

## Figures and Tables

**Figure 1 healthcare-10-01019-f001:**
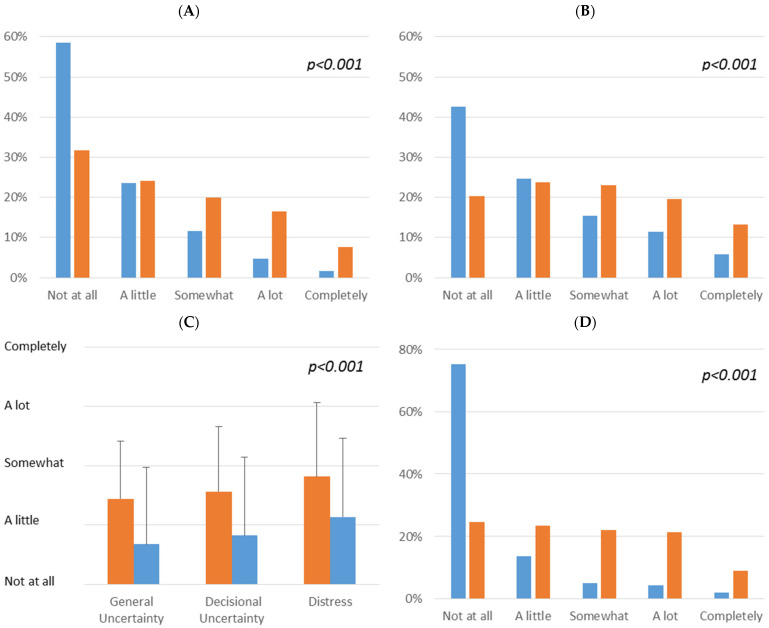
Perception of (**A**) decisional uncertainty and (**B**) distress due to uncertainty in oncology (blue) and psychiatry (orange) patients; (**C**) mean ± SD for general, treatment-related decisional uncertainty and resulting distress during own treatment decisions; (**D**) required treatment modification due to pandemic conditions from patients’ perspectives (Percentage histograms of all respondents).

**Figure 2 healthcare-10-01019-f002:**
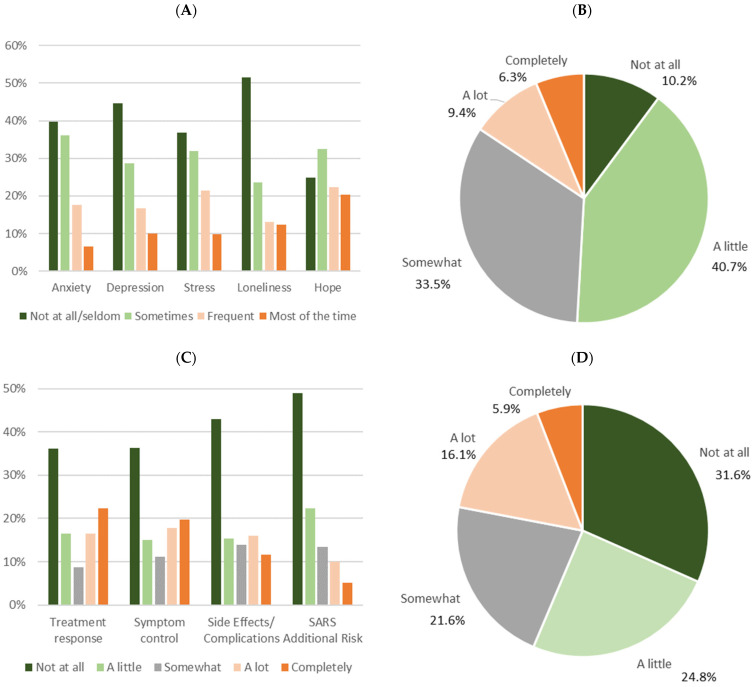
(**A**) Perception of anxiety, depression, stress, loneliness and hope within two weeks before questionnaire; (**B**) reflection of own pandemic risk during data-capture period; (**C**) modification requirement for treatment and patients’ criteria for treatment decisions; (**D**) burden due to additional risk for treatment by COVID-19 (Percentages of all respondents).

**Figure 3 healthcare-10-01019-f003:**
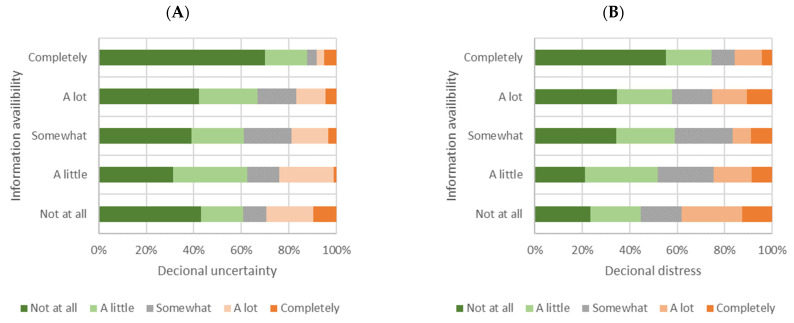
Influence of information availability on (**A**) decisional uncertainty and (**B**) decisional distress.

**Table 1 healthcare-10-01019-t001:** Prediction by nominal regression analysis for “Decisional Uncertainty” (A) Likelihood-Quotient Test for regression parameters; and (B) resulting classification.

(A)							
Effect			−2 Log Likelihood for Reduced Model		Likelihood-Quotient Tests		
					Chi²	dF	Significance
Constant Term			521.434		0.000	0	
Decision Criteria SARS Additional Risk			597.452		76.018	16	0.000
Burden Infection Risk			609.148		87.714	16	0.000
Information Availability Own Provider			592.611		71.178	16	0.000
Education Level			576.464		55.030	12	0.000
Speciality Group			586.717		65.283	4	0.000
Distance Regulations			582.071		60.638	16	0.000
Decision Criteria Side Effects/Complications			690.702		169.268	16	0.000
Decision Criteria Treatment Response			600.345		78.912	16	0.000
Pandemic Own Risk			570.796		49.363	16	0.000
**(B)**							
**Observed**	**Predicted**						
	**Not at All**	**A Little**	**Somewhat**	**A Lot**	**Completely**	**% Correct**	**% Correct with Neighbor**
Not at all	124	14	7	3	0	83.8%	93.2%
A little	31	27	5	7	0	38.6%	90.0%
Somewhat	11	6	19	5	0	46.3%	73.2%
A lot	13	6	5	22	0	47.8%	58.7%
Completely	0	0	0	0	16	100.0%	100.0%
% Total	55.8%	16.5%	11.2%	11.5%	5.0%	64.8%	85.4%

**Table 2 healthcare-10-01019-t002:** Prediction by nominal regression analysis of “Decisional Conflicts” (A) Likelihood-Quotient Test for regression parameters; and (B) resulting classification.

(A)							
Effect			−2 Log Likelihood for Reduced Model		Likelihood-Quotient Tests		
					Chi^2^	dF	Significance
Constant Term			547.700		0.000	0	
Information Availability Own Provider			575.446		27.746	16	0.034
Own Decisions			588.645		40.945	16	0.001
Decision Criteria Side Effects/Complications			585.116		37.416	16	0.002
Decision Criteria SARS Additional Risk			571.083		23.383	16	0.104
Burden Infection Risk			596.564		48.864	16	0.000
Decision Criteria Treatment Response			581.926		34.226	16	0.005
Pandemic Own Risk			580.960		33.260	16	0.007
Disease Stage			582.082		34.382	16	0.005
Age Groups			579.589		31.889	8	0.000
Education Level			575.793		28.093	12	0.005
Speciality Group			555.719		8.019	4	0.091
**(B)**							
**Observed**	**Predicted**						
	**NOT at All**	**A Little**	**Somewhat**	**A Lot**	**Completely**	**% Correct**	**% Correct with Neighbor**
Not at all	88	14	6	3	1	78.6%	91.1%
A little	25	35	8	4	1	47.9%	93.2%
Somewhat	8	13	29	2	3	52.7%	80.0%
A lot	6	9	4	22	0	53.7%	63.4%
Completely	1	1	1	0	27	90.0%	90.0%
% Total	41.2%	23.2%	15.4%	10.0%	10.3%	64.6%	85.9%

## Data Availability

The data that support the findings of this study are available on request from the corresponding author. The data are not publicly available due to privacy or ethical restrictions.

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
