# Peer review of "Decision Conflicts in Clinical Care during COVID-19: A Patient Perspective"

_healthcare, 2022, doi:10.3390/healthcare10061019_

Round 1

Reviewer 1 Report

The article has serious linguistic and grammatical errors, incorrect formatting, also in the literature. Drawings in content are truncated.

The article is supposed to be about the conflict of interest of decisions during the Covid pandemic, but neither the abstract nor the content is about it. The abstract is completely detached from the content. On the other hand, the article itself is not a bit of an analysis of the literature. The introduction is a description of admitting oncological and psychiatric patients as an attempt, no words explaining why these and not cardiological ones? The sample size is nowhere justified, and the authors conclude for the entire country. I find this too unreasonable a conclusion. The choice of the research method is also not any justification? Why was this study considered qualitative and thus statically checked? Quantitative results are hard to judge several of the figures are truncated.

Author Response

Thank you very much for your comments. Before we like to address your remarks in detail we want to express our impression of a general misunderstanding of the reviewer regarding the topic and some key elements of our investigation. The reviewer gave feedback about conflicts of interests in decisions during the pandemic. However, this quit different from decisional conflicts and decisional uncertainty as an ethical issue. Nowhere in the manuscript we address or discuss conflicts of interests. Therefore, it is not surprising that the abstract does not contain such points.

From the comment it is not clear what the reviewer means with analysis of literature. The manuscript is not intended to provide a literature review. We carefully searched for available investigations that address similar questions and referred to a larger number of these references. To address the reviewers comments we added two new references that serve as examples for decisional issues in other entities.

In the introduction we already explained why “We have chosen cancer and psychiatry patients as two groups that were assumed especially vulnerable for those problems.” We agree that other entities, such as the questioned cardiology, could be also included. However, focusing on two entities representing two very different clinical settings enabled some reflection of general phenomena and keeping the number of investigated variables in a limited complexity.

Sample size calculation in general is related to research that addresses a single primary question to compare predefined groups of patients, such as in clinical trials. However, the approach in our investigation is targeting structure finding in data which refers to different statistical settings. Power calculation is not applicable in the way as it is for group comparison. This type of analysis can only apply rough rules that are related to analytical experience, but even small groups may be used under these conditions. These experiences were considered when planning the number of participants related to the number of investigated items. When applying factorial or regression analysis it is impossible to predefine thresholds for acceptance as requirement for sample size calculation.

We agree with the comment that our investigation is not formally representative for the entire population in the country. However, the large number of sites that were included throughout Germany and the fact that they were located in regions with very different involvement in the pandemic at the time of data collection enable conclusions regarding the general occurrence of decisional issues during the pandemic. We added a paragraph in the discussion related to strengths and limitations where we addressed that our investigation is not targeting formally representative reflection throughout Germany.

Research using survey approaches is usually not targeting qualitative, but quantitative investigations. In this context we like to point out that our investigation was also planned as quantitative approach that requires statistical settings. To identify the complexity of determinants and their interlinked dependencies we even used a highly sophisticated statistical concept. For better clarification we added a sentence to the introduction. “For identification of determinants and related interlinked dependencies we have chosen a survey and multivariate analytical approach.”

We are sorry that the reviewer had difficulties to see the manuscript in a correct form. We used the journal’s template and the PDF preview as well as the word document appeared in a correct way.

The references were reformatted.

The language was carefully checked, but “serious linguistic and grammatical errors” were not found. Since the other reviewers judged the language in a different manner we kindly ask for examples that we better understand these concerns.

Reviewer 2 Report

Thank you for the opportunity to review this paper about impact of incertitude brought by the Covid on the decision the oncologic and psychiatric patients has to take.

Overall the paper is well documented with a sound research methodology and conclusions based on data. I will make a few comments on the editing of the paper. At least on my laptop (I'm using a Macbook) the references have been converted in latin numbers, I don't know if it is only my computer or something happened to the file.

In Methods please mention when exactly the data was collected.

I suggest to replace ''data capture'' (row 149) with data collection.

For Fig. 2 (the part with the histogram) please replace the colors with something more in contrast (the categories somewhat, a lot completely are almost the same when you look at the histogram), in Fig. 3 you choose the contrast and it is much more clear.

Author Response

As already mentioned in the sample description the survey wwas done between 10/2020 and 06/2021.

The wording "data capture" was replaced as suggested by "data collection".

We changed the coloring in table 2 in the same way as for figure 3.

Reviewer 3 Report

Thank you for the opportunity to review the manuscript entitled  “Decision Conflicts in Clinical Care During Covid-19: A patient perspective”.
I want to ask the authors to provide the readers with more information about the five dimensions of the Questionnaires. 
In the results section, you said that you have a sample of 540 patients. Could you please check how many are male and female, please?
I want to ask the authors to explain to readers how this study can improve clinical practice.
Could the authors complete the reference (iii)? Thank you.

Author Response

The 5 dimensions of the questionnaire were included in brackets already.

Numbers of male and female patients were added in the sample description

We have added the following paragraph to the conclusion:

Consequences recommended for pandemic patient management of clinical caregivers include

  • Assess level of uncertainty and decisional burden in patients, esp. additional effects of the pandemic
  • Consider potential modifications of decisional criteria in patients due to pandemic issues
  • Communicate own decisional uncertainty with patients to ensure shared decision making
  • Organize sufficient information management, esp. when evidence deficits occur
  • Support specific groups of vulnerable patients in handling pandemic-related additional burden

Round 2

Reviewer 1 Report

the abstract still does not reflect the content of the article or the research conducted

Author Response

The abstract was modified to better reflect the content of the investigation. Especially the methods and results are now more extensively described.